# Spatial Distribution of Atmospheric Aerosol Physicochemical Characteristics in the Russian Sector of the Arctic Ocean

**Sergey M. Sakerin [1,\*], Dmitry M. Kabanov [1], Valery I. Makarov [2], Viktor V. Pol'kin [1], Svetlana A. Popova [2], Olga V. Chankina [2], Anton O. Pochufarov [1], Vladimir F. Radionov [3] and Denis D. Rize [3]**

[1] V.E. Zuev Institute of Atmospheric Optics, Siberian Branch, Russian Academy of Sciences, Academician Zuev Square 1, 634021 Tomsk, Russia; dkab@iao.ru (D.M.K.); victor@iao.ru (V.V.P.); poa216@iao.ru (A.O.P.)

[2] V.V. Voevodsky Institute of Chemical Kinetics and Combustion, Siberian Branch, Russian Academy of Sciences, Institutskaya 3, 630090 Novosibirsk, Russia; makarov@kinetics.nsc.ru (V.I.M.); popova@kinetics.nsc.ru (S.A.P.); chankina@kinetics.nsc.ru (O.V.C.)

[3] Arctic and Antarctic Research Institute, 38 Bering Str., 199397 St. Petersburg, Russia; vradion@aari.ru (V.F.R.); denis94@yandex.ru (D.D.R.)

**\*** Correspondence: sms@iao.ru

**Abstract:** The results from studies of aerosol in the Arctic atmosphere are presented: the aerosol optical depth (AOD), the concentrations of aerosol and black carbon, as well as the chemical composition of the aerosol. The average aerosol characteristics, measured during nine expeditions (2007–2018) in the Eurasian sector of the Arctic Ocean, had been 0.068 for AOD (0.5 μm); 2.95 cm$^{-3}$ for particle number concentrations; 32.1 ng/m$^3$ for black carbon mass concentrations. Approximately two–fold decrease of the average characteristics in the eastern direction (from the Barents Sea to Chukchi Sea) is revealed in aerosol spatial distribution. The average aerosol characteristics over the Barents Sea decrease in the northern direction: black carbon concentrations by a factor of 1.5; particle concentrations by a factor of 3.7. These features of the spatial distribution are caused mainly by changes in the content of fine aerosol, namely: by outflows of smokes from forest fires and anthropogenic aerosol. We considered separately the measurements of aerosol characteristics during two expeditions in 2019: in the north of the Barents Sea (April) and along the Northern Sea Route (July–September). In the second expedition the average aerosol characteristics turned out to be larger than multiyear values: AOD reached 0.36, particle concentration up to 8.6 cm$^{-3}$, and black carbon concentration up to 179 ng/m$^3$. The increased aerosol content was affected by frequent outflows of smoke from forest fires. The main (99%) contribution to the elemental composition of aerosol in the study regions was due to Ca, K, Fe, Zn, Br, Ni, Cu, Mn, and Sr. The spatial distribution of the chemical composition of aerosols was analogous to that of microphysical characteristics. The lowest concentrations of organic and elemental carbon (OC, EC) and of most elements are observed in April in the north of the Barents Sea, and the maximal concentrations in Far East seas and in the south of the Barents Sea. The average contents of carbon in aerosol over seas of the Asian sector of the Arctic Ocean are OC = 629 ng/m$^3$, EC = 47 ng/m$^3$.

**Keywords:** Arctic Ocean; aerosol; black carbon; organic and elemental carbon; elemental composition

## 1. Introduction

Environmental studies in the Arctic zone have acquired priority in the last decade. The increased attention to the Arctic stems from the development of this area, the largest dynamics of climate characteristics, and the vulnerability of the environment to these changes [1]. An important role in the formation processes of climate and ecological environmental state is played by atmospheric aerosol [2]. The Arctic has few local sources of aerosol emissions; however, the physicochemical composition of the Arctic atmosphere is impacted strongly by outflows of different pollutants from Eurasian and North American midlatitudes [3,4]. Here we primarily mean the long–range transports of submicron (fine) aerosol of natural and anthropogenic origins, the lifetime of which reaches a week or longer. During transport process, the physicochemical transformation of aerosol occurs in the changing meteorological conditions and stratifications of the atmosphere. Moreover, the Arctic troposphere at the beginning of polar day is often characterized by temperature inversions, leading to the accumulation of pollutants, called the Arctic haze phenomenon [5–7].

In addition to its own effect on climate and ecologic environmental state, aerosol influences the adjacent climate–forming factors [2,8,9]. First, aerosol participates in the processes of cloud formation and transformation of cloud radiation characteristics (reflection, absorption). Second, the deposition of absorbing aerosol leads to decrease in the albedo of snow–covered underlying surface, typical for polar regions. The reviewing works [7,10] summarized comprehensively the studies of atmospheric aerosol in polar regions over a few decades. However, the results, obtained in the Russian sector of the Arctic, are presented only partly and only until 2015.

At present, regular measurements of aerosol characteristics in the Eurasian sector of the Arctic are carried out at three stations: in Barentsburg (Spitsbergen Archipelago), in the region of Tiksi, and in the "Cape Baranov" (Severnaya Zemlya Archipelago). Measured characteristics include aerosol optical depth (AOD) of the atmosphere, aerosol and black carbon concentrations, as well as the chemical composition of aerosol samples. The results obtained serve as a basis for determining seasonal variations in aerosol characteristics, sources of aerosol emissions and long–range transport pathways, absorbing properties, and radiation effects [11–17]. In addition to observations at polar stations, expedition measurements of aerosol characteristics are carried out every year in different regions of the Arctic Ocean and North Atlantic [17–23].

Special attention is devoted recently to studying the absorbing components of aerosol (black and brown carbon), which are formed during combustion of different fuel and biomass types. The instrumental measurements of absorption with the use of aethalometers fail to identify uniquely the absorbing substances in aerosol composition. Therefore, we additionally (or alternatively) collected samples on filters for determining subsequently the morphology and chemical composition of aerosol by different methods of the microscopy, spectral analysis, chromatography, and electron paramagnetic resonance (EPR) spectroscopy [24–28].

One of the unsolved tasks is to determine the spatially average distribution of characteristics of aerosol over the Arctic Ocean, the formation of which is impacted by the outflows of aerosols of different types from midlatitudes. Results of individual expeditions answer the question as to what physicochemical characteristics of aerosol can be in specific weather conditions. And to determine of the regularities of spatial distribution, multiyear uniform measurements are required. Additionally, determination of the regularities of spatial distribution require multiyear uniform measurements. Data accumulation during nine expeditions allowed us to carry out a statistical generalization and to obtain for the first time the estimates of the distribution of aerosol characteristics in Eurasian sector of the Arctic Ocean (30° E–10° W). In addition to multiyear average data, in this work we also discuss the measurements of aerosol physicochemical characteristics during two new expeditions *TransArctic–2019* (cruises of RV *Akademik Tryoshnikov* and RV *Professor Multanovskiy*).

## 2. Characterization of Expedition Measurements

In the period from 2007 to 2018, we carried out measurements of aerosol characteristics from aboard RVs in nine Arctic expeditions: the list of expeditions is shown in Figure 1. In 2019 we continued expedition studies of atmospheric aerosol over seas of the Arctic Ocean. The measurements of aerosol characteristics were carried out at the stages 1 and 4 of the complex expedition "TransArctic–2019": "TA–2019/St. 1" and "TA–2019/St. 4".

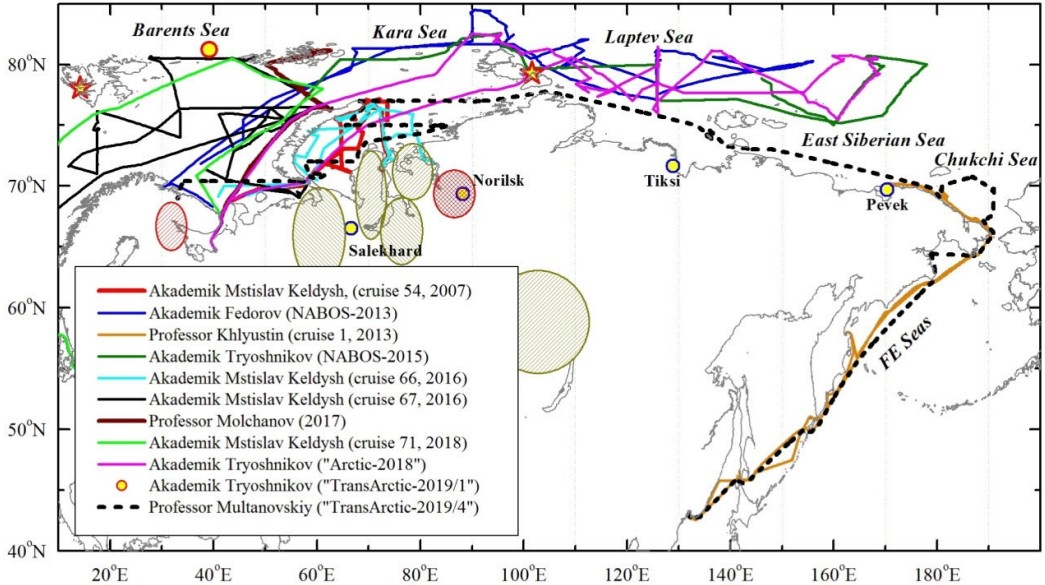

**Figure 1.** Routes of Arctic expeditions (circle indicates the region of drift in ice during "TA–2019/Stage 1"; stars indicate the polar stations "Barentsburg" and "Cape Baranov"; shaded areas indicate the main areas of extraction of mineral resources (gas, oil, etc.) production in the north of Russia.

As in the previous expeditions [13,19,29], the measurements were performed using the same set of instruments, comprising particle counter AZ–10 [30], aethalometer MDA [31], and portable sun photometer SPM [32]. Based on data from these instruments, for every hour of measurements we determined the following characteristics:

- Mass concentration of absorbing substance (black carbon, $M_{BC}$) in the aerosol composition;
- Number concentration of aerosol particles with radii $r = 0.15$–$5$ μm ($N_A$);
- Volume particle distribution functions $(dV_i/dR_i) = 4/3 \cdot \pi \cdot r_i^3 \cdot dN_i$, where $dN_i$ and $r_i$ are the number concentrations of particles and average radii for the $i$th size interval;
- Volumes of particles in the fine and coarse fractions of aerosol ($V_f$ and $V_c$), which were calculated from the formula $V = \Sigma \; 4/3 \cdot \pi \cdot r_i^3 \cdot dN_i$ for the radius ranges 0.15–0.5 and 0.5–5 μm, respectively;
- AOD of the atmosphere $\tau^a(\lambda)$ in the wavelength range of 0.34–2.14 μm and parameters of the *Ångström formula* $\alpha$, $\beta$;
- Fine and coarse components of AOD (the coarse component $\tau^c$ was found from minimal AOD values in the spectral range of 1.24–2.14 μm, and the fine component was estimated for the wavelength of 0.5 μm as the residual depth [33]: $\tau_{0.5}^{f} = \tau_{0.5}^{a} - \tau^{c}$).

We will give a clarification about the set of aerosol characteristics chosen. It is well known that the particles in fine and coarse aerosol fractions differ in the natures of their origin, processes of transformation in the atmosphere, propagation distances, and lifetimes. Therefore, in studying aerosol variations, the behavior of each fraction is expedient to consider separately, both in the

near–surface layer and in the atmospheric depth. In the previous works (e.g., [19,23]) we analyzed the individual variations of number concentrations in two particle fractions ($N_f$ and $N_c$), while in the current case the particle volumes $V_f$ and $V_c$ will be analyzed. This change is explained by two circumstances.

First, the distribution functions of the number concentrations are characterized by a very rapid decrease with the growing particle size (the exponent is about 3 in the Junge power–law distribution). For this reason, in the $N_f$ or $N_c$ values the concentrations of the smallest particles (near the left boundary of the size range) predominate, while the contribution of larger particles is barely manifested. That is, the use of $N_f$ (or $N_c$) gives a distorted idea about the behavior of the whole fraction.

Second, in analysis it is better to employ precisely particle volumes because they are invariant with respect to aerosol mass concentration: $M \sim \rho \cdot V$, where $\rho$ is the density of aerosol substance. For instance, the authors of works [34,35] use $\rho = 1.5$ g/cm$^3$ in the calculations of the mass concentration of continental aerosol. Here, we restrict ourselves only to the calculations of particle volumes $V_f$ and $V_c$ in view of uncertain $\rho$ values for Arctic aerosol and, even more so, for individual fractions.

In addition to instrumented measurements of aerosol characteristics, a vortical air blower ERSTEVAK EVL 22/11 was used to collect aerosol samples on *Whatman®* and AFA–KhA–20 filters for a subsequent analysis under the laboratory conditions (see [13,29] for more detail). The fiberglass *Whatman®* filters were used to determine the concentrations of organic (OC) and elemental carbons (EC) with the help of the method of reaction gas chromatography [36]. The samples, collected on acetyl cellulose AFA–KhA–20 filters, we used to determine aerosol mass concentration (PM) by the gravimetric method, as well as the elemental composition of aerosol by the method of Synchrotron radiation X–ray fluorescence (SXRF) analysis [37].

At the 1st stage of the expedition ("TA–2019/St. 1") the aerosol characteristics were measured from April 7 to 29 in the region of drift of RV *Akademik Tryoshnikov* in ice. The measurement pavilion was on ice floe 400 m from the vessel. The vessel, frozen in ice, drifted in the northern part of the Barents Sea within the following coordinates: 80°19′–81°30′ N; 38°24′–39°54′ E. The geographic position of this region, as well as those of the routes of other marine expeditions, discussed in the present paper, are shown in Figure 1.

In the second expedition ("TA–2019/St. 4") the measurements were carried out onboard RV *Professor Multanovskiy* from 28 July to 7 September along the route from Vladivostok to Murmansk. The statistical characteristics of aerosol were calculated for individual Arctic seas and for the whole Eurasian sector of the Arctic Ocean. Due to their shortage, data of measurements in the East Siberian and Chukchi Seas were combined into a single subset. In addition, average characteristics for three Far East seas (FE Seas: Sea of Japan, Sea of Okhotsk, and Bering Sea) were calculated jointly (without dividing with respect to regions).

Ship–based measurements of near–surface aerosol characteristics are subject to the effect of unfavorable weather conditions (spray, precipitation, fog) and technogenic sources (smoke from chimney, polluted air from ventilation systems). Measurements and sampling were terminated when such effects occurred. In addition, false measurements were eliminated at the stage of initial data processing. Tables 1 and 2 presents the amount of refined data, obtained in the "TransArctic–2019" expedition, as well as the total number of days of measurements in nine preceding expeditions, in which our studies were carried out. We note that the number of days of measurements in separate expeditions and seas is rounded off (toward the larger values) to give the integer number of days. Therefore, the sum of days in separate seas (expeditions) differs a little from the total number of days, presented in the last column and lower row of Table 2.

**Table 1.** Number of hours and days of measurements aerosol characteristics ($\tau^a$–$N_A$–$M_{BC}$) in expeditions "TransArctic–2019": Stage 1 (7–28 April) and Stage 4 (28 July–7 September).

| | Stage 1: Drift in the ice RV *Akademik Tryoshnikov* | | Stage 4: Cruise of the RV *Professor Multanovskiy* | | | |
|---|---|---|---|---|---|---|
| | North of the Barents Sea | South of the Barents Sea | Kara Sea | Laptev Sea | East Siberian and Chukchi Sea | Far East Seas |
| Number of hours | 38–495–0 | 5–21–35 | 2–46–87 | 0–0–0 | 8–74–117 | 12–131–168 |
| Number of days | 9–23–0 | 2–3–3 | 1–7–7 | 0–0–0 | 2–8–8 | 4–17–17 |

**Table 2.** Number of days of measurements aerosol characteristics ($\tau^a$–$N_A$–$M_{BC}$) in various Arctic seas and expeditions 2007–2018.

| RVs Expeditions | Barents Sea (BS) | Kara Sea (KS) | Laptev Sea (LS) | East Siberian and Chukchi Sea (ESCS) | Total for the Expedition |
|---|---|---|---|---|---|
| *Akademik Mstislav Keldysh* (cruise 54, September–October 2007) | 0–5–5 | 0–27–27 | 0–0–0 | 0–0–0 | 0–30–30 |
| *Akademik Fedorov* (NABOS–2013, August–September) | 0–9–9 | 0–9–9 | 1–19–19 | 1–5–5 | 2–27–27 |
| *Professor Khlyustin* (cruise 1, July– September 2013) | 0–0–0 | 0–0–0 | 0–0–0 | 0–13–12 | 0–13–12 |
| *Akademik Tryoshnikov* (NABOS–2015, August–October) | 0–11–11 | 0–22–21 | 1–12–12 | 0–13–13 | 1–51–50 |
| *Akademik Mstislav Keldysh* (cruise 66, July–August 2016) | 2–5–5 | 18–35–34 | 0–0–0 | 0–0–0 | 20–38–37 |
| *Akademik Mstislav Keldysh* (cruise 67, August–October 2016) | 12–39–38 | 0–0–0 | 0–0–0 | 0–0–0 | 12–39–38 |
| *Professor Molchanov* (July 2017) | 7–12–12 | 0–0–0 | 0–0–0 | 0–0–0 | 7–12–12 |
| *Akademik Mstislav Keldysh* (cruise 71, July–August 2018) | 1–7–7 | 0–0–0 | 0–0–0 | 0–0–0 | 1–7–7 |
| *Akademik Tryoshnikov* ("Arctic–2018", August–September) | 1–4–4 | 1–8–8 | 4–22–21 | 2–12–12 | 8–40–39 |
| Total by seas (2007–2018) | 23–92–91 | 19–101–99 | 6–53–52 | 3–43–42 | 51–257–252 |

## 3. Discussion of Results

In Section 3.1 we considered the specific features of the spatially average distribution of microphysical and optical characteristics of aerosol over the Arctic Ocean according to the data of our multiyear (2007–2018) studies (the expeditions are listed in Figure 1). As compared to analogous analysis, presented in [23], we corrected initial subsets of data, associated with the refinement of the boundaries of the Kara Sea and separate regions of the Barents Sea. As a consequence of this correction, the average values of $N_A$ and $M_{BC}$ changed a little in the Barents Sea, as well as $N_A$ in the Kara Sea and in the south of the Barents Sea.

In two subsections below we will discuss the results from complex aerosol studies in two new expeditions "TransArctic–2019": aerosol optical and microphysical characteristics in Section 3.2, and results of analysis of chemical composition of aerosol samples in Section 3.3.

### 3.1. Average Spatial Distribution of Aerosol Optical and Microphysical Characteristics over the Arctic Ocean

The aerosol characteristics, measured during marine expeditions, depend primarily on specific synoptic conditions and atmospheric circulations in short periods of measurements in a particular region. To identify the specific features of the average geographic distribution of aerosol, not

one–time cycles or expeditions, but long–term (multiyear) observations, are required in the whole variety of synoptic and meteorological conditions that are characteristic for each region.

Measurements in nine Arctic expeditions permitted us to estimate the specific features of the spatial distribution of aerosol in the Russian sector of the Arctic Ocean, namely, the latitudinal variations in aerosol characteristics over the Barents Sea and longitudinal (from west to east) change over other seas: the Barents Sea (BS), Kara Sea (KS), Laptev Sea (LS), and East Siberian Sea together with Chukchi Sea (ESCS).

For the Barents Sea basin, strong spatial variations in aerosol and black carbon concentrations were already noted earlier [23]. The highest aerosol content is observed in the southern part of the Barents Sea; the $N_A$ and $M_{BC}$ concentrations gradually decrease with the distance from continent. To estimate the quantitative differences, we calculated the average aerosol characteristics for three subregions: the southern part of the Barents Sea (<71° N; abbreviated as SBS), the middle part (71–80° N; MBS), and the northern part (>80° N; NBS).

From Figure 2 (see left part) it is clearly seen that the aerosol characteristics decrease from southern toward northern part of the Barents Sea: $M_{BC}$ decreases by a factor of 1.5; $N_A$ by a factor of 3.7; $V_f$ by a factor of 5.1, $V_c$ by a factor of 8.6. The high aerosol and black carbon contents in the south of the Barents Sea are explained by the proximity to densely populated and industrially developed regions of Scandinavia and Pomorie (Murmansk and Arkhangelsk regions), as well as by heavier ship traffic in this region. The northward decrease in aerosol concentrations is favored by (aside from increasing distance from continent) the growth in ice–covered sea surface and the decrease in air temperature, solar insolation, and content of aerosol–forming substances.

From the general considerations, a decrease in aerosol characteristics in the northern direction should also be expected over other Arctic seas; however, no quantitative estimates could still be obtained due to the scarcity of measurements at different latitudes. For this same reason (data shortage) we do not present AOD characteristics in three parts of the Barents Sea, although higher atmospheric turbidities are also manifested in the southern part [23].

Comparison of average aerosol characteristics over separate seas (see Table 3 and right part of Figure 2) showed a general decreasing tendency of concentrations in the eastern direction (from BS toward ESCS): $M_{BC}$ and $N_A$ decreased by a factor of 1.9; $V_f$ by a factor of 1.5, and $V_c$ by a factor of 2.1. Only particle volumes ($V_f$ and $V_c$) over the Kara Sea show large values and deviate from this regularity (see below for reasons).

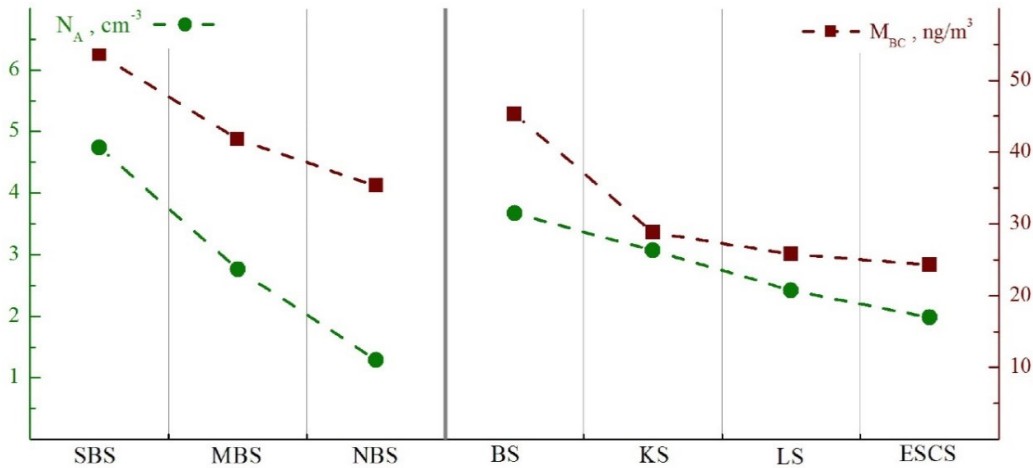

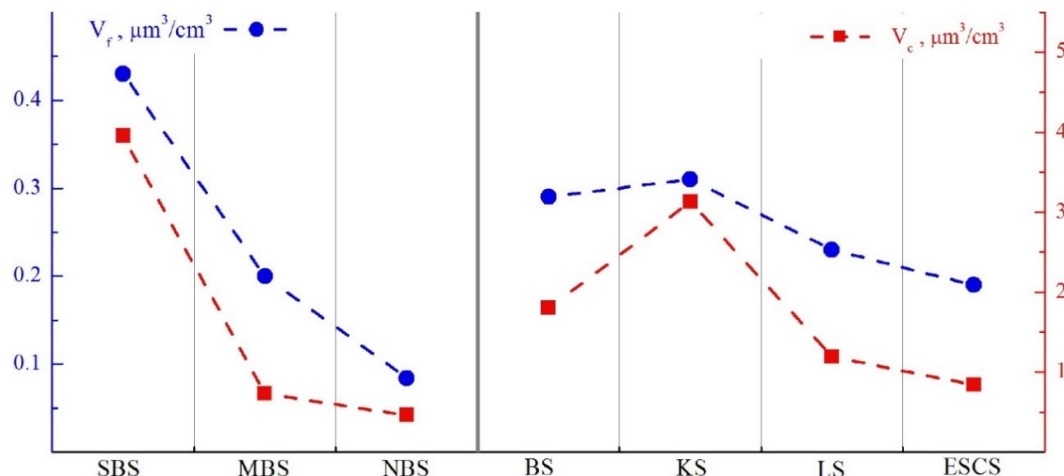

**Figure 2.** Multiyear (2007–2018) average concentrations of aerosol and black carbon in different regions of the Arctic Ocean: in three parts of the Barents Sea (SBS, MBS, NBS; left side), and over the Arctic seas (BS, KS, LS, ESCS; right side).

**Table 3.** Average values and standard deviations (±SD) of aerosol characteristics over the Arctic seas using data from multiyear measurements in the 2007–2018 expeditions.

| Characteristics | Barents Sea (BS) | Kara Sea (KS) | Laptev Sea (LS) | ESC Seas (ESCS) | Arctic Ocean |
|---|---|---|---|---|---|
| $M_{BC}$, ng/m$^3$ | 45.3 ± 67.5 | 28.8 ± 27.9 | 25.8 ± 34.0 | 24.3 ± 36.3 | 32.1 ± 44.1 |
| $N_A$, cm$^{-3}$ | 3.67 ± 3.14 | 3.07 ± 3.45 | 2.42 ± 3.48 | 1.98 ± 2.80 | 2.95 ± 3.36 |
| $V_f$, μm$^3$/cm$^3$ | 0.29 ± 0.29 | 0.31 ± 0.43 | 0.23 ± 0.36 | 0.19 ± 0.29 | 0.28 ± 0.37 |
| $V_c$, μm$^3$/cm$^3$ | 1.80 ± 3.78 | 3.13 ± 6.15 | 1.19 ± 2.21 | 0.84 ± 1.36 | 2.14 ± 4.69 |
| $\tau_{0.5}^{a}$ | 0.080 ± 0.051 | 0.057 ± 0.046 | 0.082 ± 0.077 | 0.037 ± 0.012 | 0.068 ± 0.052 |
| $\tau_{0.5}^{f}$ | 0.051 ± 0.048 | 0.024 ± 0.023 | 0.063 ± 0.072 | 0.018 ± 0.006 | 0.040 ± 0.044 |
| $\tau^{c}(\approx\beta)$ | 0.029 ± 0.021 | 0.033 ± 0.025 | 0.019 ± 0.014 | 0.019 ± 0.011 | 0.028 ± 0.021 |
| $\alpha$ | 1.00 ± 0.46 | 0.35 ± 0.23 | 0.84 ± 0.37 | 0.74 ± 0.47 | 0.72 ± 0.48 |

The statistical estimates according to Student's *t*–test showed that the average values of most aerosol characteristics over Arctic seas differ with the 0.95 confidence probability. Exceptions are minor (insignificant) differences between aerosol characteristics in just two cases: in the average $M_{BC}$ values between LS and ESCS regions, and in $V_f$ between BS and KS regions. If we consider the whole totality of aerosol characteristics, the differences between regions and the decreasing tendency in the eastern direction are statistically significant with respect to a few parameters.

More detailed idea about the spatial differences in aerosol disperse composition can be gained from volume particle distribution functions *dV/dr* (Figure 3a). In the range of the smallest (*r* < 0.35 μm) particles, high *dV/dr* values are observed over the Barents Sea and, especially, in its southern part (SBS). Over other seas, *dV/dr* values decrease in the eastern direction, in an analogous way to the particle volumes $V_f$.

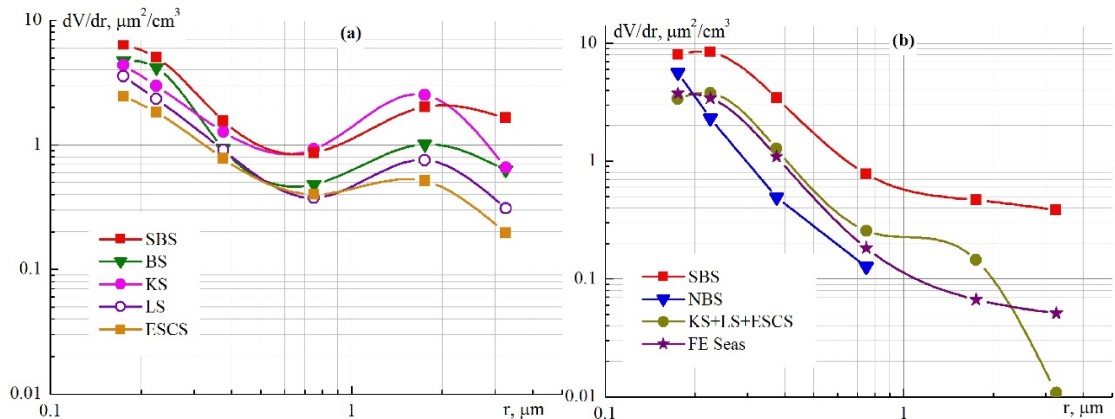

**Figure 3.** Average volume particle distribution functions (*dV/dr*): (**a**) multiyear (2007–2018) data; (**b**) in expedition "TransArctic–2019".

In the size range of coarse aerosol, the salient feature of the Kara Sea had been well manifested: the average *dV/dr* values are about the same as in the south of the Barents Sea, and much (a factor of 2.5–5) larger than over other Arctic seas. Considering that the content of fine aerosol over the Kara Sea is also quite high (see $V_f$ in Figure 2), the effect of outflows of pollutants from oil and gas production plants in the north of the Western Siberia could be speculated. However, this version is in a bad agreement with the average $M_{BC}$ and $N_A$ values over the Kara Sea (see upper part of Figure 2), which do not distort the decreasing tendency in the eastern direction.

We will clarify what can be responsible for the increased content of coarse aerosol over the Kara Sea. Long–range transports of continental aerosol influence the specific features of the spatial distribution of concentrations of fine aerosol; while the content of coarse aerosol is determined mainly by local or regional factors. The production of coarse marine aerosol is known to depend on wind speed and sea roughness [2]. In Arctic regions, the additional effect is due to the ice cover of sea surface (relative ice area), reducing the generation of marine aerosol. Therefore, there are grounds to believe that the large $V_c$ and *dV/dr* values over the Kara Sea, as well as in the south of the Barents sea, were due to the smaller ice cover and larger (on the average) wind speeds than in other seas.

In the spatial AOD distribution over the Arctic Ocean (see lower part of Table 3) we can also see that the average values decreased from the Barents Sea toward ESC Seas: $\tau_{0.5}^a$ decreases by more than a factor of 2. Mainly, this AOD change was due to the fine component; while, the average $\tau^c$ values varied in a small range of values 0.019–0.033.

The tendency of the longitudinal AOD decrease is distorted over the Laptev Sea. The AOD value was larger in this region due to the situation of powerful outflow of smoke from forest fires on 26 August 2018 [23]. After this anomalous situation was eliminated, the average AOD over the Laptev Sea decreased by a factor of 1.5 (from $\tau_{0.5}^a$ = 0.082, as in Table 3, to $\tau_{0.5}^a$ = 0.053) and became intermediate between the neighboring seas (Figure 4a).

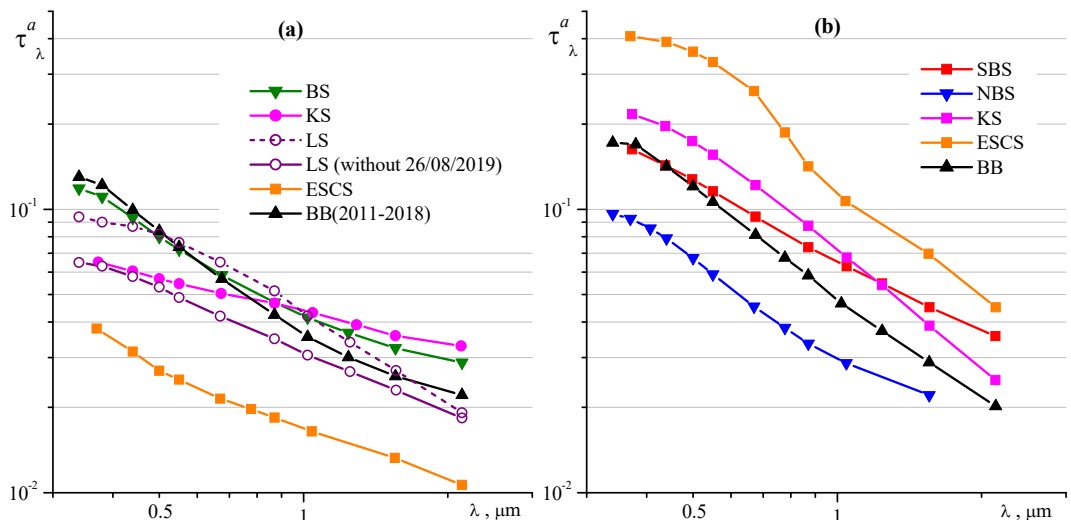

**Figure 4.** Average spectral dependences of AOD: (**a**) multiyear data in different regions of the Eurasian sector of the Arctic (BB/2011–2018 is Barentsburg); (**b**) measurements in 2019.

The average spectral dependences $\tau^a(\lambda)$, presented in Figure 4a, make it possible to refine that AOD decrease in the eastern direction is manifested only in the shortwave part of the spectrum: the largest AOD values over the Barents Sea, the middle level over the Kara and Laptev Seas (without the situation on 26 August 2018), and low values in ESC Seas. However, when we pass to the IR range ($\lambda > 1$ μm), this order changes: AOD values over the Kara Sea become larger than over the Barents Sea. We also note that the wavelength dependence $\tau^a(\lambda)$ over the Kara Sea is the flattest, and the *Ångström exponent* ($\alpha = 0.35$) is a factor of 2–3 smaller than in other regions ($\alpha = 0.74$–1). This feature is because $\tau^c$ is large over the Kara Sea and makes the predominant contribution to AOD. This is consistent with the largest particle volumes of coarse aerosol ($V_c$) in the near–surface atmosphere of the Kara Sea (see the discussion above).

For a comparison, Figure 4a presents the average wavelength dependence of AOD at the Arctic station in Barentsburg (BB/2011–2018; [13]), located on Spitsbergen Archipelago. As compared to the Kara Sea and the neighboring Barents Sea, the AOD values in Barentsburg are the largest in the shortwave wavelength range and become smaller in the IR wavelength range. This behavior of $\tau^a(\lambda)$ can be explained by higher content of coarse aerosol over the seas at smaller content of fine aerosol. For this reason, the wavelength dependences of AOD, observed in Barentsburg, exhibit a fast decrease with the growing wavelength and, correspondingly, a higher *Ångström exponent* ($\alpha \approx 1.3$), as compared with marine regions ($\alpha = 0.35$–1).

We will consider the possible causes explaining why the aerosol characteristics decrease from the Barents Sea to ESC Seas. The main pollution sources of the atmosphere over the Arctic Ocean are known to be: (a) regular outflows of anthropogenic aerosol and other types of aerosol from continents (predominantly from industrially developed regions of Europe), and (b) episodic, but stronger outflows of smoke from forest fires in boreal zone. For absorbing aerosol, comparative analyses of the influence of these sources on the Arctic atmosphere were considered in many works (see, e.g., [4,14,18,38]).

These results indicate that the anthropogenic aerosol sources, located in Europe, are nevertheless predominant. As we move away from Scandinavia in the eastern direction, the contents of fine aerosol and black carbon over the Arctic Ocean decrease, and the effect of outflows (including smoke from fires) from sparsely populated regions of Siberia turns out to be less significant and does not compensate for this decrease. Of course, this conclusion cannot be extended to certain periods of measurements, when the effect of smoke outflows may predominate and distort the pattern of the average spatial distribution of aerosol. A characteristic example is the results from summer

measurements of aerosol characteristics during "TransArctic–2019" expedition (see discussion below).

As regards to how aerosol characteristics are affected by smoke from forest and vegetation burning, we will make two clarifications. First, not only Siberia, but also other midlatitude regions, namely, Far East, Europe, and North America, are the sources of smoke aerosol [8,12,39–42]. Distances of long–range transports of smoke aerosol from these regions to Eurasian sector of the Arctic have comparable values: of the order of 2000–4000 km.

Second, the outflows of smoke plumes lead to considerable AOD increases (such as in situation on 26 August 2018); however, they are not always accompanied by simultaneous increases in aerosol and black carbon contents in the near–surface atmospheric layer. Although, smoke outflows are ultimately followed by smoke diffusion and subsidence to the lower layers. That is, the average levels of aerosol and black carbon concentrations in the near–surface level of the Arctic atmosphere also increase, to a certain degree.

If we do not consider the specific features of the spatial distribution, the main characteristics of aerosol in the Eurasian sector of the Arctic Ocean have the following average values (see last column in Table 3): $\tau_{0.5}^a$ = 0.068; $N_A$ = 2.95 cm$^{-3}$, and $M_{BC}$ = 32.0 ng/m$^3$.

The average black carbon concentrations, which we obtained in the Eurasian sector of the Arctic Ocean, agree with measurements in [21] during 2017: 76.6 ng/m$^3$ over the Norwegian Sea; from 36 to 46 ng/m$^3$ over the Barents Sea; about 23 ng/m$^3$ over the Laptev Sea. Over the Kara Sea, the authors of the work noted the increase in the black carbon concentration by more than 100 ng/m$^3$ due to episodes with outflow of smoke from forest fires and associated gas combustion products from north of Siberia. Approximately the same data were obtained during an expedition in 2015 [20]: from the Barents toward Laptev Sea, the black carbon concentrations varied from 10 to 116 ng/m$^3$, the average being 34 ng/m$^3$. The authors of work [22] noted strong variations in the black carbon content over the Kara Sea and in the south of the Barents Sea: together with background concentrations (less than 30 ng/m$^3$), over the two–week period they recorded a few extreme concentrations (up to 160 and 360 ng/m$^3$), caused by outflows of pollutants from the direction of oil and gas production regions.

### 3.2. Optical and Microphysical Characteristics of Aerosol in the Expeditions "TransArctic–2019"

*Stage 1.* Figure 5 illustrates the variations of aerosol characteristics in the region of drift of RV *Akademik Tryoshnikov* in ice. In the period of this expedition (April 2019) there were mainly low values of AOD and aerosol concentrations: $\tau_{0.5}^a$ < 0.08, $N_A$ < 2.5 cm$^{-3}$. High aerosol content was recorded in three cases: AOD reached 0.12 on 7 and 22 April, while on April 26 the concentration $N_A$ increased to 4.7 cm$^{-3}$. Data from trajectory analysis (https://ready.arl.noaa.gov/HYSPLIT.php) and the foci of temperature anomalies (https://worldview.earthdata.nasa.gov) showed that in the first two cases, the air masses were supplied from Eastern Siberia: on 7 April from the north of Yakutia; on 22 April from Taymyr and southwest of Yakutia (Figure 6), where there are large oil and gas production centers (Chayanda, Talakan field, etc.). Correspondingly, the sources of pollutants could be emissions from boilers or stove heating systems in the north settlements; on 22 April they could additionally be from associated gas combustion. The aerosol concentration maximum, observed on 26 April, was likely due to the Arctic haze phenomenon, because air was transported only from the Arctic Ocean basin.

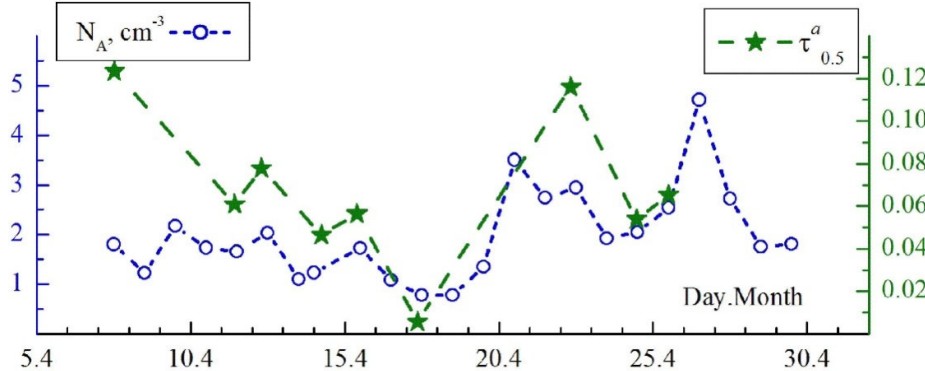

**Figure 5.** Variations in daily average concentrations $N_A$ and AOD in the expedition of RV *Akademik Tryoshnikov* (drift in ice).

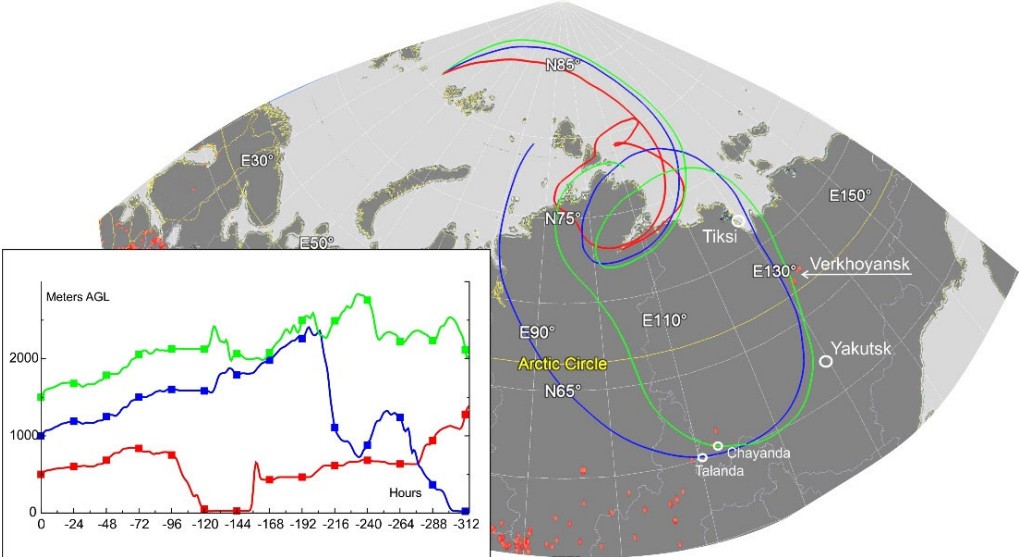

**Figure 6.** Back trajectories and heights above ground level (AGL) of air mass motion to the region of drift of RV *Akademik Tryoshnikov* in ice on 22 April 2019.

The AOD and $N_A$ maxima, indicated above, impacted also the average aerosol characteristics, presented in Table 4. The average AOD values over the entire period of measurements ($\tau_{0.5}^a = 0.067$) turned out to be a little larger than in the previous expeditions in this region ($\tau_{0.5}^a = 0.051$), but smaller than the averages in Barentsburg [13] (see Figure 4b). The aerosol concentration in the near–surface atmospheric layer also exceeded the multiyear average value: $N_A = 2.01$ cm$^{-3}$ in April 2019 versus $N_A = 1.29$ cm$^{-3}$ in the previous expeditions (Figure 7). The average particle distribution function (lower curve in Figure 3b) is characterized by a rapid drop with the growing particle size, taking near–zero values in the size range $r > 1$ µm. That is, there were almost no particles of coarse aerosol in the period of measurements on ice floe.

**Table 4.** Average (±SD) characteristics of aerosol in the expeditions "TransArctic–2019".

| Characteristics | Stage 1 | Stage 4 | | | | |
|---|---|---|---|---|---|---|
| | NBS | SBS | KS | LS | ESCS | FE Seas |
| $M_{BC}$, ng/m³ | – | 179 ± 133 | 39 ± 30 | 34 ± 31 | 74.1 ± 56.5 | 108 ± 93 |
| $N_A$, cm⁻³ | 2.01 ± 1.10 | 8.62 ± 2.65 | 2.83 ± 1.60 | – | 4.17 ± 3.52 | 3.26 ± 3.64 |
| $V_f$, µm³/cm³ | 0.16 ± 0.09 | 0.85 ± 0.26 | 0.27 ± 0.16 | – | 0.38 ± 0.34 | 0.30 ± 0.35 |
| $\tau_{0.5}^{a}$ | 0.067 ± 0.036 | 0.128 ± 0.021 | 0.174 ± 0.013 | – | 0.360 ± 0.188 | 0.264 ± 0.077 |
| $\tau_{0.5}^{f}$ | 0.045 ± 0.031 | 0.092 ± 0.029 | 0.149 ± 0.005 | – | 0.315 ± 0.174 | 0.217 ± 0.071 |
| $\tau^{c}(\approx\beta)$ | 0.022 ± 0.011 | 0.036 ± 0.012 | 0.025 ± 0.008 | – | 0.045 ± 0.014 | 0.047 ± 0.021 |
| $\alpha$ | 1.23 ± 0.38 | 0.99 ± 0.09 | 1.19 ± 0.06 | – | 1.14 ± 0.12 | 1.25 ± 0.13 |

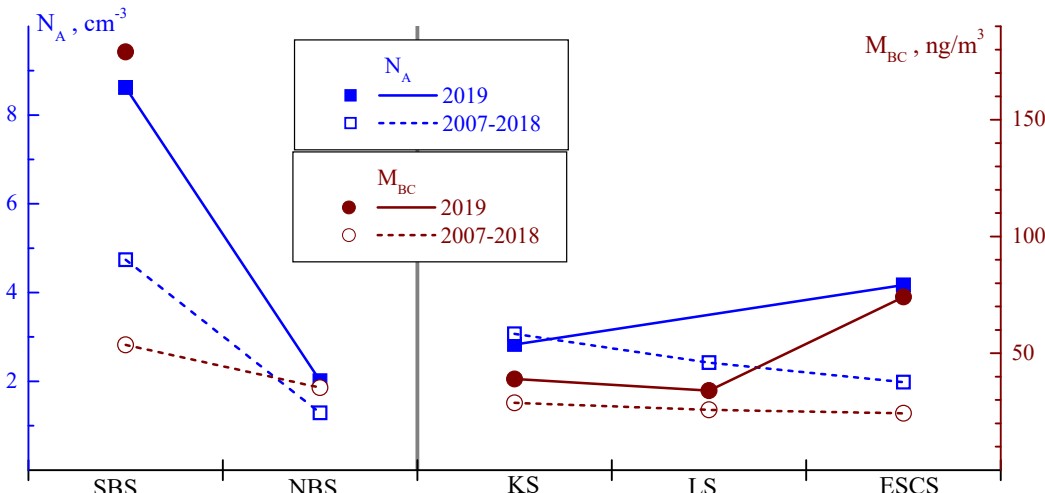

**Figure 7.** Average concentrations $N_A$ and $M_{BC}$ in five regions of the Arctic Ocean (NBS, SBS, KS, LS, ESCS): in expedition "TransArctic–2019" and multiyear (2007–2018) data.

*Stage 4*. Aerosol characteristics varied in a much wider variability range in cruise of RV *Professor Multanovskiy* (Figure 8). The diurnally average aerosol and black carbon concentrations were high mainly near continents: on 27–29 July in the Sea of Japan (receding from Vladivostok), on 8 August on roadstead of port of Anadyr, on 18 August in the Long Strait (ESCS), and on 6/7 September in the south of the Barents Sea (approaching Murmansk). The maximum of aerosol concentration on 18 August ($N_A$ = 8.60 cm⁻³) was due to smoke haze and, as such, was not accompanied by increase in black carbon concentrations. The increases in the concentrations $N_A$ and $M_{BC}$ on 6/7 September were due outflow of smoke aerosol from the Baltic States (Figure 9).

The largest AOD value ($\tau_{0.5}^{a}$ > 0.5) was recorded on 17 August in the Chukchi Sea. Data from trajectory analysis indicate that smoke from forest fires in the north of Yakutia and Western Siberia were transported at that time (Figure 9). High AOD values on 28 July and 2 August were also associated with outflows of smoke aerosol to the Far Eastern Seas (FE Seas) from Siberia and north of China.

A preliminary analysis of measurements with AZ–10 counter revealed that aerosol concentrations in the size range of the largest particles ($r$ > 1.5 µm) were strongly underestimated. The calculated $dV/dr$ values turned out to be about an order smaller than usual (see Figure 3b). The particle counter was tested after expedition; the electronic system of instrument was found to operate incorrectly in two last channels (particle size ranges). This defect did not distort measurements (counting) of concentrations in channels of submicron particles and influenced weakly the total concentration $N_A$. In view of these circumstances, the concentrations of coarse aerosol ($N_c$, $V_c$) were excluded from analysis, and only $N_A$ and $V_f$ were considered.

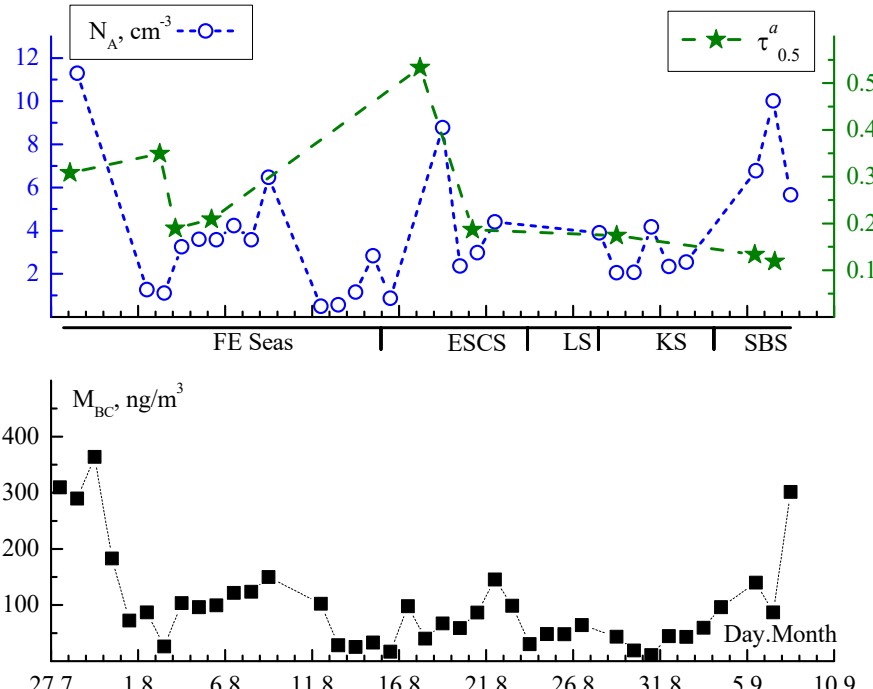

**Figure 8.** Variations in daily average concentrations $N_A$, $M_{BC}$, and in AOD in expedition of RV *Professor Multanovskiy*.

Comparison of aerosol characteristics along the route of RV *Professor Multanovskiy* with multiyear data from previous expeditions gave the following results. The average aerosol characteristics over individual Arctic seas in 2019 varied in the ranges (see Table 4 and Figure 7): $\tau_{0.5}^a$ = 0.128–0.360, $N_A$ = 2.83–8.62 cm⁻³, $V_f$ = 0.27–0.85 μm³/cm³, and $M_{BC}$ = 34–179 ng/m³. That is, multiyear data were exceeded by most characteristics in most regions. The average AOD had been larger than the multiyear values by a factor of 1.8–10; near–surface concentrations exceeded those by a factor of 1.3–3.3. The only exceptions were comparable aerosol concentrations over the Kara Sea: $N_A$ = 2.83 cm⁻³, $V_f$ = 0.27 μm³/cm³ in 2019 versus multiyear values $N_A$ = 3.07 cm⁻³, $V_f$ = 0.31 μm³/cm³.

In summer 2019 the aerosol content increased in the size range of submicron particles (see $\tau_{0.5}^f$ and $V_f$) under the influence of outflows of smoke from forest fires to the regions of measurements. The enrichment of the Arctic atmosphere by fine aerosol was well manifested in the particle distribution functions $dV/dr$ in the region SBS (Figure 3b), in the wavelength dependences $\tau^a(\lambda)$ in ESCS (Figure 4b), as well as in the increased *Ångström exponent* $\alpha$. The average Ångström exponents over the Arctic Seas were in the range $\alpha$ = 0.35–1 in the previous expeditions and increased to $\alpha$ = 1–1.14 in 2019.

We also note that aerosol characteristics over ESC seas in the period of the expedition turned out to be comparable with, or even higher than, those over FE (Japan, Okhotsk, and Bering) Seas, which experience a stronger effect of continental outflows from China and Far East of Russia.

Thus, a salient feature of the expedition on RV *Professor Multanovskiy* is frequent outflows of smoke from forest fires to the regions of measurements. As a consequence of outflows of smoke plumes, in three cases we recorded AOD maxima (0.32–0.54) and observed higher level of aerosol and black carbon contents in the near–surface atmospheric layer.

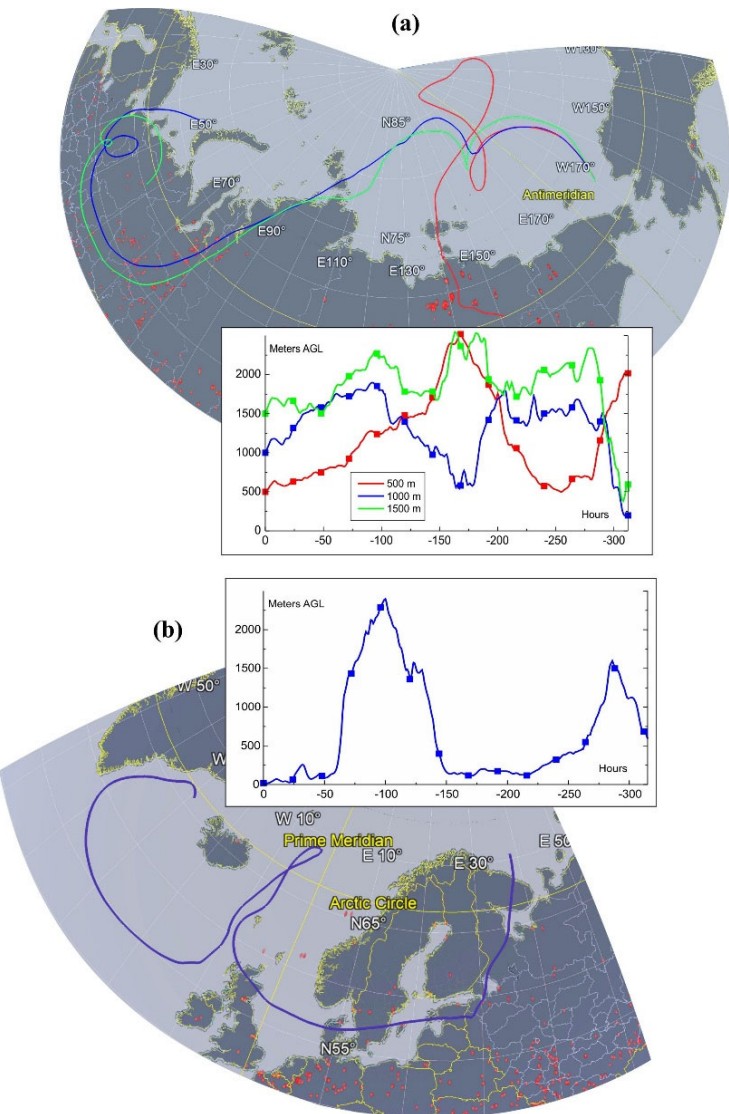

**Figure 9.** Back trajectories and heights above ground level (AGL) of air mass motion to the regions of measurements and fire centers: (**a**) on 17 August in the Chukchi Sea; (**b**) on 6 September toward the south of the Barents Sea (near Murmansk) to the height of 20 m.

### 3.3. Chemical Composition of Aerosol Samples along the Route of "TransArctic–2019" Expeditions

Figure 10 shows the dynamics of concentrations of organic (OC) and elemental (EC) carbons in the aerosol composition, measured during two Arctic expeditions. Despite the fact that aerosol samples were collected on filters for a relatively long period of time (material was accumulated for 2–3 days), the characteristics obtained turned out to be quite variable. The variability range reaches a factor of seven for OC and EC concentrations, and a little smaller (a factor of 3–5) for aerosol mass concentration PM. We note that the main role in the variations of aerosol characteristics is played not by spatial inhomogeneities, but by synoptic oscillations with periods of a few days, associated with changes of air masses. This is evident from measurements in remote region of drift of RV *Akademik Tryoshnikov* in ice (Figure 10a): the EC concentration varied from 11 to 72 ng/m³, the OC concentration from 181 to 1374 ng/m³, and PM from 1.6 to 4.9 μg/m³.

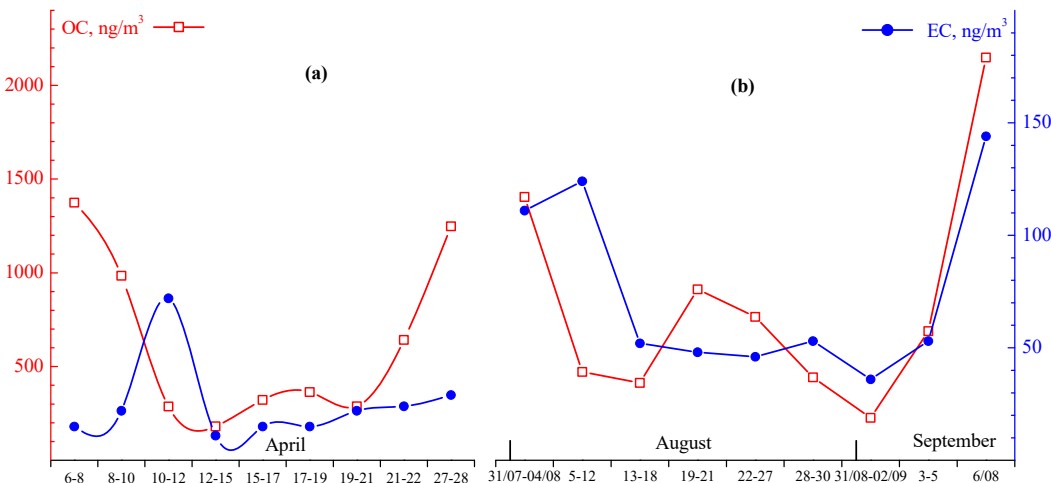

**Figure 10.** Variations in OC and EC concentrations in expedition of RV *Akademik Tryoshnikov* (**a**) and *Professor Multanovskiy* (**b**).

In the period of this expedition, the maximal concentrations were observed on 10–12 April for EC, and on 6–8 April and 27–28 April for OC. The non–coincidence of the maxima of OC and EC and the absence of correlation (R = 0.13) between their variations indicates that EC and OC have different emission sources. The main sources of EC are the combustion of different fuel types and forest fires. Emission sources of organic carbon are more numerous and diverse: vegetation waste products, biomass burning, and processes of catalytic and photochemical oxidation of gas–phase hydrocarbons in the atmosphere [43,44].

Back trajectory analysis of air mass motion (https://ready.arl.noaa.gov/HYSPLIT.php) gave no unique answer about what the sources of EC and OC are and why their maxima occur in different times (6–8 April and 10–12 April). We only note that for the majority of time of taking the corresponding aerosol samples, air was transported from the territory of the Arctic Ocean and snow–covered Arctic coast, where there are no significant aerosol sources. In the middle of the sampling periods (7 and 11 April), there were also long–range transports of air from southwest of Siberia and oil and gas production areas (Figure 11a). The temperature anomalies in the southwest of Siberia indicate combustion centers. That is, the sources of EC and OC emissions could be vegetation burning products, associated gas combustion as well as smoke from stove heating systems and boilers.

During the period of maximum OC concentrations on 27–28 April, the trajectories of air movement to the measurement site also passed over the Arctic Ocean. Additionally, only in the evening of 27 April, air masses were transported from the territory of Gyda Peninsula (Figure 11b), where large oil and gas fields are being developed. A probable OC source in this case could be emissions or combustion of hydrocarbon.

In the second expedition (RV *Professor Multanovskiy*) the largest carbon concentrations in aerosol samples were recorded on September 6 in the south of the Barents Sea (Figure 10b): OC = 2159 ng/m³, EC = 144 ng/m³. As was already indicated above (see Section 3.2), in this period there were high concentrations $N_A$ and $M_{BC}$, caused by outflow of smoke and anthropogenic aerosol from the territory of Baltic countries (see Figures 8 and 9).

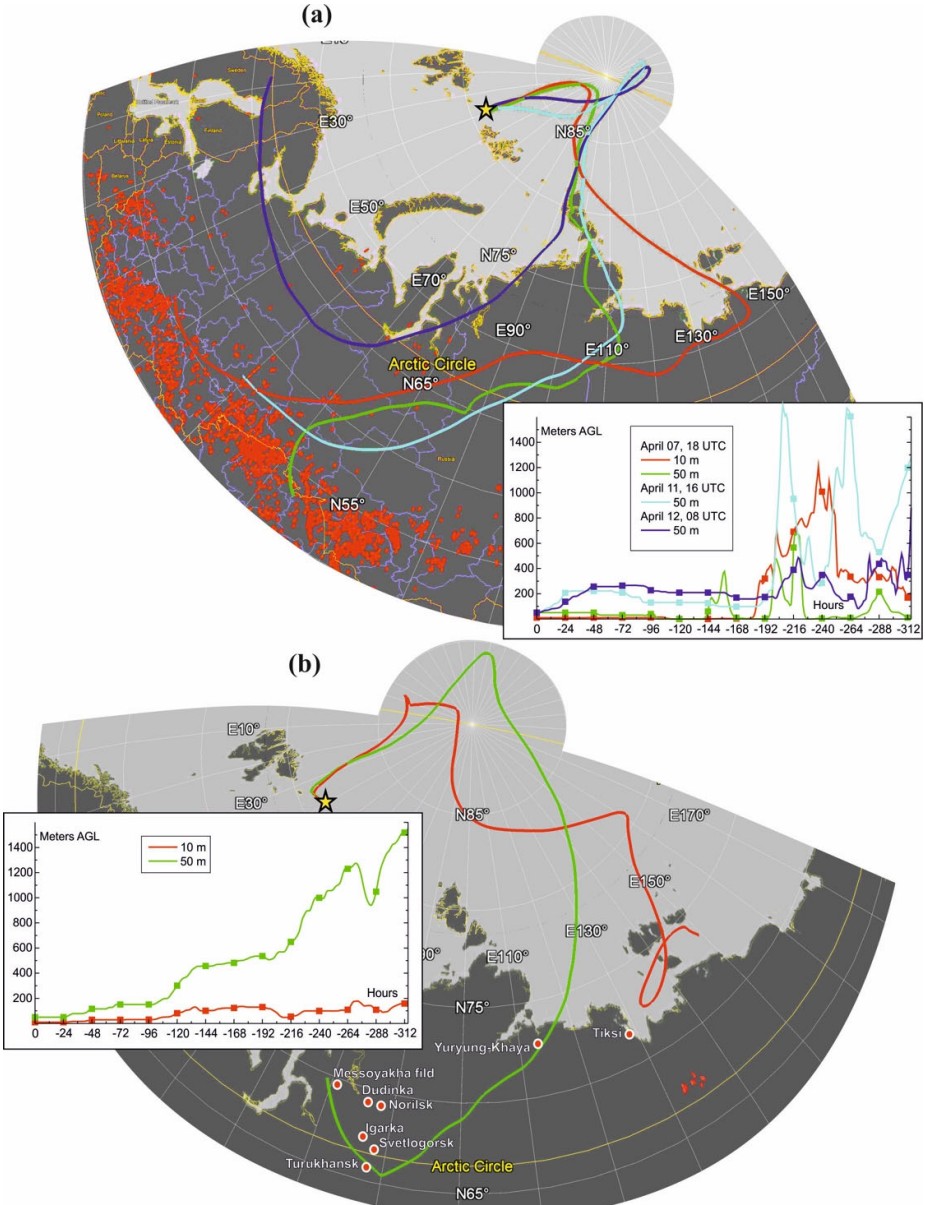

**Figure 11.** Back trajectories and heights above ground level (AGL) of air mass motion to the regions of measurements and fire centers: (**a**) on 7 and 11 April; (**b**) on 27–28 April.

Somewhat smaller maxima of carbon concentrations were recorded in Far East seas (FE seas): on 31 July–4 August during sailing around of Kamchatka Peninsula (OC = 1446 ng/m³, EC = 111 ng/m³) and on 5–12 August (EC = 124 ng/m³) in Anadyr Bay. Proximity to Kamchatka coasts influenced the high OC and EC contents in the first sample (on 31 July–4 August). The maximum of the EC concentration on 5–12 August was due to anthropogenic effect of the port of Anadyr during long–term stop of the vessel on roadstead.

Table 5 presents the average characteristics of aerosol samples in separate regions of two expeditions: Stage 1, RV *Akademik Tryoshnikov*; Stage 4, RV *Professor Multanovskiy*. From these data it follows that the Barents Sea exhibits highly inhomogeneous aerosol characteristics: low OC, EC, and PM concentrations in the northern part (in the region of drift in ice) and maximal concentrations in the south (SBS). The average OC, EC, and PM concentrations in the south of the Barents Sea turned out to be comparable to, or larger than, those in FE seas. The characteristics of aerosol samples in the Asian sector of the Arctic Ocean (KS, LS, ESCS; collectively called ASAO) are intermediate between

the data in NBS and SBS and agree well with measurements at polar station "*Cape Baranov*", located between the Kara and Laptev Seas [13]: OC = 509 ng/m$^3$; EC = 65 ng/m$^3$; PM = 9.1 μg/m$^3$.

**Table 5.** Average (±SD) OC, EC, and PM concentrations in separate regions of the "TransArctic–2019" expedition (*n* is the number of aerosol samples).

| | Stage 1 | | | Stage 4 | | | |
|---|---|---|---|---|---|---|---|
| Characteristics | NBS (*n* = 9) | SBS (*n* = 2) | KS (*n* = 2) | LS (*n* = 1) | ESCS (*n* = 2) | ASAO | FE Seas (*n* = 2) |
| *OC*, ng/m$^3$ | 632 ± 458 | 1460 ± 989 | 420 ± 146 | 833 | 735 ± 337 | 629 ± 269 | 999 ± 633 |
| *EC*, ng/m$^3$ | 25 ± 19 | 98 ± 64 | 45 ± 12 | 46 | 50 ± 2.9 | 47 ± 8 | 118 ± 9.1 |
| *PM*, μg/m$^3$ | 3.3 ± 1.8 | 21.9 ± 17 | 7 ± 0.1 | 13 | 10.8 ± 2.5 | 9.7 ± 2.9 | 11.5 ± 1.8 |

Table 6 presents the average concentrations of elements in atmospheric aerosol in the study regions. The main (99%) contribution to the elemental composition of aerosol is due to Ca, K, Fe, Zn, Br, Ni, Cu, Mn, and Sr, which refer to elements of both natural and anthropogenic origins. The first three elements (Ca, K, Fe) account for 68% of the total content of the studied elements in the most remote region of drift in ice (NBS), and their contribution is about 90% in other regions.

The spatial distribution of elemental composition exhibited the same features, indicated for other aerosol characteristics: the lowest concentrations of the most elements (except Br, Cu, Rb) are observed in the high–latitude region NBC; the maximal concentrations in the FE seas and/or in the south of the Barents Sea (SBS). The average concentrations of elements in the region NBC are a factor of 3.6–11 smaller than over FE seas and SBS. The concentrations of elements over the seas of the Asian sector of the Arctic Ocean (KS, LS, ESCS) are in the middle of the total range and differ much less among themselves. Therefore, Table 6 presents them jointly, for the entire ASAO sector.

Comparison with elemental composition of continental aerosol [13,45] showed that the (decreasing) order in the distribution of concentrations of elements in the Arctic atmosphere is nearly the same; but the concentrations are smaller in value. The largest (about an order of magnitude or more) difference in the concentrations is characteristic for Ca, K, Fe, Ti, Sr, Pb, Zr, and Mo. A smaller (several–fold) difference is found for Zn, Mn, Co, Ni, Se, and Nb. The Cu concentration shows comparable values to continental aerosol; while Br is more abundant (especially in NBS).

Depending on its remoteness from continents and predominant air mass transports, the atmosphere over ocean is enriched to some degree by aerosols with different chemical compositions (organic, anthropogenic, smoke, etc.). The high concentrations of the most elements over SBS and FE seas are caused by their proximity to continental sources, located in industrially developed and more densely populated regions of northern Europe and Far East. The smaller concentrations of elements over the seas of the Asian sector of the Arctic Ocean indicate that the effect of the northern territories of Siberia is not as significant. This is explained by the weakness of anthropogenic sources in these regions (despite the development of mineral resource extraction) and long snow–covered period, inhibiting the emission of aerosol material from the underlying surface.

**Table 6.** Average (±SD) concentrations of elements in aerosol composition (ng/m$^3$) during "TransArctic–2019" expeditions (*n* is the number of samples).

| Elements | NBS (*n* = 9) | ASAO (*n* = 5) | SBS (*n* = 2) | FE Seas (*n* = 2) |
|---|---|---|---|---|
| Ca | 154 ± 140 | 414 ± 272 | 808 ± 534 | 965 ± 708 |
| K | 88 ± 47 | 294 ± 178 | 824 ± 478 | 742 ± 745 |
| Fe | 47 ± 34 | 308 ± 287 | 144 ± 85 | 490 ± 32 |
| Zn | 30 ± 13 | 30 ± 9.1 | n.d. | 47 ± 34 |
| Br | 79 ± 44 | 20 ± 9.1 | 46 ± 22 | 32 ± 28 |
| Ti | 11 ± 4.0 | 25 ± 13 | 55 ± 55 | 70 ± 97 |
| Cu | 12 ± 8.0 | 3.6 ± 1.4 | 52 ± 17 | 28 ± 7.1 |
| Mn | 1.6 ± 0.80 | 10 ± 8.8 | 19 ± 8.0 | 14 ± 6.2 |

| | | | | |
|---|---|---|---|---|
| Sr | 0.96 ± 0.60 | 3.0 ± 1.4 | 3.7 ± 1.9 | 7.1 ± 6.0 |
| Co | 0.31 ± 0.16 | 1.8 ± 1.2 | 2.4 ± 1.3 | 3.4 ± 0.1 |
| Pb | 0.66 ± 0.91 | 2.0 ± 1.1 | 2.2 ± 2.1 | 3.9 ± 2.0 |
| Ni | 0.99 ± 0.46 | 1.5 ± 1.9 | n.d. | 6.0 ± 7.2 |
| Se | 0.10 ± 0.05 | 0.31 ± 0.23 | 0.13 ± 0.09 | 0.80 ± 0.55 |
| Rb | 0.27 ± 0.21 | 0.22 ± 0.13 | 0.37 ± 0.23 | 0.39 ± 0.29 |
| Zr | 0.13 ± 0.18 | 0.31 ± 0.35 | 0.32 ± 0.39 | 0.74 ± 0.87 |
| Nb | 0.05 ± 0.03 | 0.07 ± 0.03 | 0.24 ± 0.17 | 0.21 ± 0.17 |
| Mo | 0.08 ± 0.06 | 0.16 ± 0.03 | 0.22 ± 0.20 | 0.29 ± 0.17 |

n.d. is the undefined element.

The above–mentioned distribution of concentrations over the regions is distorted for a few elements. This is primarily true for bromine, the concentration of which in the high–latitude region NBS turned out to be much larger than in other regions. In addition, the lowest Cu and Rb concentrations were found not in the region NBS, but over the seas of the Asian sector.

We note once more the wide variability range of elemental composition of aerosol within a single region (like for the OC and EC concentrations), indicating the strong dependence on the air mass type. Even in the most remote and clean region of drift in ice (NBS), the concentrations of elements in separate samples differed by a few times, and by more than an order of magnitude for Ca, Fe, Cu, Pb, and Zr. The largest concentrations of Cu, Pb, Zr (on 27–28 April) and Ca, Fe (on 12–15 April) turned out to be close to the level of the average concentrations in the regions SBS or FE seas. In the first case, the maximal Cu, Pb, and Zr concentrations coincided in time with high content of organic carbon in aerosol (see Figure 10a). In the second case, the maxima of the Ca and Fe concentrations were not accompanied by increases in other aerosol characteristics. That is, the microphysical characteristics of aerosol and its chemical composition vary with the change in air mass significantly, but not always in coordination.

In this paper we do not consider the interrelations in variations of physicochemical characteristics of aerosol in the Arctic atmosphere; they will be analyzed in a separate paper on a larger statistical material (using data from a few expeditions).

In conclusion, we will compare the elemental composition of aerosol in the "TransArctic–2019" expeditions (NBS, SBS, ASAO, FE seas) with data from our previous measurements in the high–latitude regions: at polar station *"Cape Baranov"* during spring 2018 [13] and in the 71st cruise of RV *Akademik Mstislav Keldysh* in the North Atlantic (average concentrations at the 57–72° N latitudes) [29]. From Figure 12 it can be seen that the concentrations of most elements (except for Br, and partly for Cu, Rb) were grouped according to three types of the regions: the lowest content of elements is observed in the region of drift in ice (NBS); the middle level in the Asian sector of the Arctic Ocean (ASAO), North Atlantic, and at the station *"Cape Baranov"*; maximal level in the regions of FE seas and/or SBS. This distribution of elemental composition of aerosol is consistent with the impact degree of continental outflows, which enrich the Arctic atmosphere by additional aerosol.

The main salient feature of the distribution of elements over regions concerns bromine, the concentration of which turned out to be the largest in the high–latitude region NBS. The comparisons of elemental composition of aerosol showed that increased bromine content is also characteristic for samples collected during spring at the station *"Cape Baranov"*. A common feature of these regions is that the majority of their territories was covered by ice and snow in the period of measurements. The high bromine content in polar regions during the spring period was noted by many authors [46–48]. The recent studies [48] indicate that the source of bromine emissions is the new sea ice and surface snow.

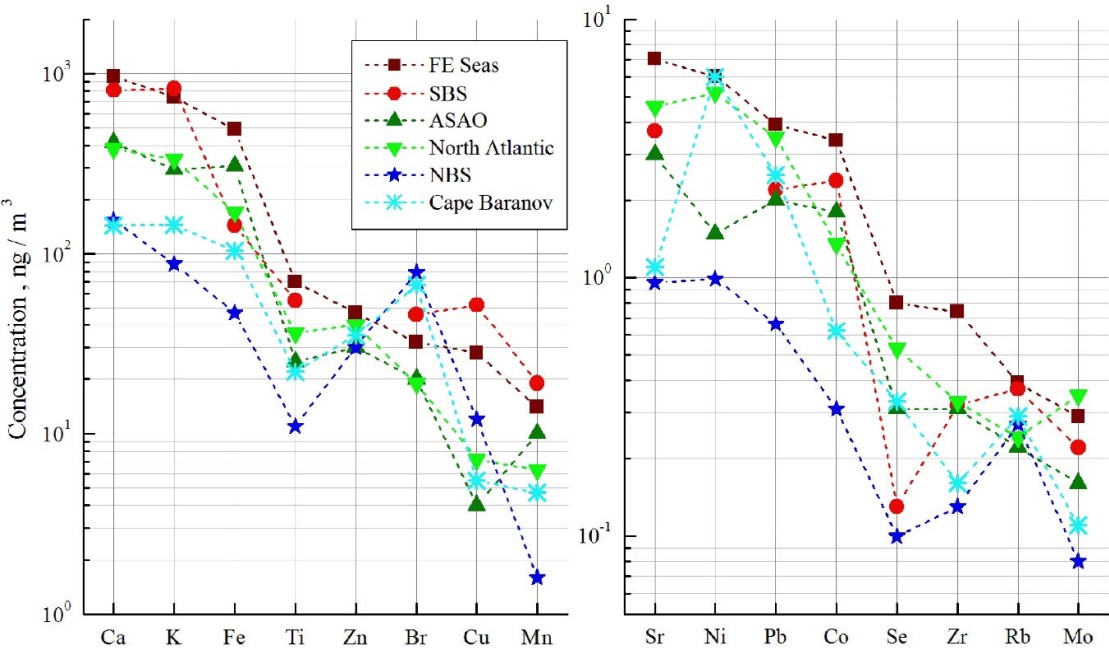

**Figure 12.** Comparison of the average elemental composition of aerosol in different regions: in "*TransArctic–2019*" expeditions (NBS, SBS, ASAO, and FE seas); at "*Cape Baranov*" station [13]; in the North Atlantic (71st cruise of RV *Akademik Mstislav Keldysh*) [29].

## 4. Conclusions

We analyzed the data from multiyear expedition studies of aerosol in the Arctic Ocean and results from two expeditions in 2019: in high–latitude part of the Barents Sea (drift of RV *Akademik Tryoshnikov* in ice) and along the Northern Sea Route (RV *Professor Multanovskiy*). A statistical generalization of results from nine previous expeditions (2007–2018) showed that aerosol characteristics in the Eurasian sector of the Arctic Ocean, on the average, have the following values: $\tau_{0.5}^a$ = 0.068; $\tau_{0.5}^f$ = 0.04; $\tau^c$ = 0.028; $\alpha$ = 0.72; $N_A$ = 2.95 cm$^{-3}$; $V_f$ = 0.28 μm³/cm³; $V_c$ = 2.14 μm³/cm³; $M_{BC}$ = 32.1 ng/m³.

Two regularities have been manifested in the spatial distribution of aerosol over the Arctic Ocean. The first regularity concerns the Barents Sea basin, over which the average aerosol characteristics decrease in the northern direction: $M_{BC}$ decreases by a factor of 1.5, $N_A$ by a factor of 3.7, $V_f$ by a factor of 5.1, and $V_c$ by a factor of 8.6. The increased aerosol and black carbon contents in the southern part of the Barents Sea (near Scandinavia) were caused by the effect of outflows of pollutants from the continent and heavy ship traffic in this region.

The second regularity is the longitudinal distribution of aerosol in the Eurasian sector of the Arctic Ocean, which is characterized by about twofold decrease of the average characteristics in the eastern direction. From the Barents toward ESC Seas, $\tau_{0.5}^a$ decreases from 0.080 to 0.037, $N_A$ from 3.67 to 1.98 cm$^{-3}$, $V_f$ from 0.29 to 0.19 μm³/cm³, and $M_{BC}$ from 45.3 to 24.3 ng/m³. These features of the spatial distribution were mainly caused by variations in the content of fine aerosol ($\tau_{0.5}^f$, $V_f$). The decrease in aerosol characteristics from west toward east indicates that outflows of fine aerosol of different types to the Arctic atmosphere are overall larger from Europe than from the Asian part of Russia. The differences in the average aerosol characteristics over individual Arctic seas and the tendency in the eastern direction are statistically significant with respect to a few parameters.

The content of coarse aerosol is determined by local or regional factors (wind speed and sea ice cover). Higher (3.1–3.8 μm³/cm³) $V_c$ values were obtained in the south of the Barents Sea and over the Kara Sea; $V_c$ < 1.2 μm³/cm³ in other regions.

It is noted that AOD and aerosol characteristics in the near–surface atmospheric layer respond differently to outflows of smoke and other pollutants from midlatitudes. Independent of transport altitude, smoke plumes always increase AOD of the atmosphere. Ground–level aerosol and black

carbon concentrations increase in the cases when outflows are accompanied by air subsidence in the region of measurements. However, the Arctic atmosphere is ultimately enriched by additional aerosol in each air mass transport from continents. As a consequence, the average regional level of aerosol and black carbon concentrations in the near–surface layer also increases with a certain delay.

This difference in the responses of AOD and near–surface concentrations was well manifested in results from expedition studies in 2019. During two expeditions "TransArctic–2019", in five cases the AOD increases were directly associated with outflows of pollutants (mainly smokes) from midlatitudes; while the near–surface concentrations increased due to outflow in just one case on September 6/7. Nonetheless, not only AOD, but also near–surface characteristics of aerosol in 2019 turned out to be larger than the multiyear average values. In the period of summer expedition of RV *Professor Multanovskiy* the average characteristics of aerosol in separate regions of the Arctic Ocean varied in the ranges: $\tau_{0.5}^a$ = 0.128–0.360, $N_A$ = 2.83–8.62 см⁻³, $V_f$ = 0.27–0.85 μm³/cm³, and $M_{BC}$ = 34–179 ng/m³. In the April expedition (drift of RV *Akademik Tryoshnikov* in ice) the average characteristics of aerosol had been: $\tau_{0.5}^a$ = 0.067, $N_A ≈ 2$ cm⁻³, and $V_f$ = 0.16 μm³/cm³.

The chemical composition of aerosol samples (concentrations of OC, EC, and elements) is also characterized by strong interdiurnal variations, caused by change in air mass. Even within a single region, the variations of carbon concentrations in aerosol composition reach a factor of seven, and more than an order of magnitude for certain elements. The largest OC and EC concentrations in the "TransArctic–2019" expedition were recorded in the nearby impact zone of continental aerosol, namely: in the south of the Barents Sea during outflow of aerosol pollutants from the direction of Baltic countries (OC = 2159 ng/m³, EC = 144 ng/m³); near the Kamchatka coasts (OC = 1446 ng/m³, EC = 111 ng/m³), and in the Gulf of Anadyr (EC = 124 ng/m³).

The main (99%) contribution to elemental composition of aerosol in the study regions is due to Ca, K, Fe, Zn, Br, Ni, Cu, Mn, and Sr. The first three elements (Ca, K, Fe) contribute 68% in the most remote region of drift in ice (NBS) and about 90% in all other regions.

The spatial distribution of the chemical composition of aerosol exhibited the regularities analogous to the distribution of optical and microphysical characteristics. The lowest average concentrations of carbon (OC = 632 ng/m³, EC = 25 ng/m³) and of most elements (except Br, Cu, and Rb) are observed in the high–latitude region NBC; the maximal concentrations (OC = 1460 ng/m³, EC = 98 ng/m³) in the Far East seas and in the south of the Barents Sea. The chemical composition of aerosol over seas of the Asian sector of the Arctic Ocean differ less significantly, with the average carbon content in this sector (ASAO) being OC = 629 ng/m³, EC = 47 ng/m³. This distribution of concentrations of OC, EC, and elements (minimal, middle, and maximal levels) is consistent with the strength of continental aerosol sources and predominant circulations of air masses, which enrich the Arctic atmosphere by additional aerosol.

In contrast to the most elements, the largest average concentration of bromine was observed in the high–latitude region NBS, consistent with the data from samples, collected in spring at the polar station *"Cape Baranov"* [13]. The source of increased bromine content in these regions seems to be the emissions from sea ice.

**Author Contributions:** Conceptualization, S.M.S.; Organization of expeditionary measurements, V.F.R.; Data curation, D.M.K.; Expeditionary measurements, A.O.P. and D.D.R.; Physicochemical analysis of samples, V.I.M., O.V.C. and S.A.P.; Data processing and formal analysis—D.M.K., S.A.P., V.I.M. and V.V.P.; Writing—original draft, S.M.S.; Writing separate sections, V.V.P. and S.A.P.; Writing—review and editing, S.M.S. All authors have read and agreed to the published version of the manuscript.

**Funding:** This work was supported by the Ministry of Science and Higher Education of the Russian Federation (Project No. AAAA–A17–117021310142–5).

**Acknowledgments:** The authors thank their colleagues, who participated in measurements and in preparation of instrumentation, i.e., to O.N. Izosimova, Vas.V. Polkin, A.P. Rostov, V.P. Shmargunov, S.A. Terpugova, S.A. Turchinovich and P.N. Zenkova; also thanks go to the developers of the NASA Worldview application (https://worldview.earthdata.nasa.gov), part of the NASA Earth Observing System Data and Information System (EOSDIS), for providing the possibility to use the imagery of thermal anomalies, and NOAA Air

Resources Laboratory (https://ready.arl.noaa.gov/HYSPLIT.php) for providing the possibility to use the trajectory model.

**Conflicts of Interest:** The authors declare no conflict of interest.

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
