# Peer review of "Spatial Distribution of Atmospheric Aerosol Physicochemical Characteristics in the Russian Sector of the Arctic Ocean"

_atmosphere, doi:10.3390/atmos11111170_

Round 1
Reviewer 1 Report
This revised manuscript has responded to my opinions.
Reviewer 2 Report
The authors have revised the manuscript to mostly address the concerns raised in my first review of this manuscript. Their response notes that there is not enough data to quantify north south gradients for the Kara Sea. Their revised discussion of the back-trajectory analysis in Figure 11 addresses concerns raised in the review.
Regarding proper English, the authors ask for "specific paragraph(s) or a sentence(s)" that are not clear. There are too many instances for me to list, which is why I still recommend editorial review for proper English.
I leave it up to the editorial staff to determine how to proceed.
Reviewer 3 Report
I am satisfied with the corrections made and wish future professional success to the authors.
This manuscript is a resubmission of an earlier submission. The following is a list of the peer review reports and author responses from that submission.
Round 1
Reviewer 1 Report
The authors monitor the aerosol physicochemical characteristics in the Russian sector of the Arctic Ocean. Results showed the spatial distribution variations of atmospheric AOD, NA, Vf, Vc, and MBC. It also provides the OC, EC, and elemental composition of aerosol from two expeditions in 2019. It is suitable to publish in the atmosphere.
There are a few comments as follows for the authors' reference:
- Line 121. Check the unit of density.
- Section 3.1. Were they significant differences in the data among these locations? I think authors should take a statistic test.
- Line, 273. Check the letters. It is not the English alphabet.
- Section 3.3. I suggest that authors should compare the EC and MBC from their measurement results. I think they are valuable information for readerships.
Reviewer 2 Report
Overall comments: This manuscript needs to be reviewed for proper English grammar prior to publication. There are numerous grammatical errors throughout the manuscript that make it difficult to comprehend what the authors are trying to say.
The new information presented in the manuscript appears to be limited to the two 2019 expeditions, with only very minor changes in the analysis associated with refinement of the boundaries of the Kara Sea and separate regions of the Barents Sea of the previous nine campaigns that was published in Russian in 2019 [Radionov, V.F. et al., 2019]. I question whether Section 3.1 and the discussion associated with Figure 2 and Table 2 are necessary. It is also not clear whether the AOD results shown in Table 2 are from the Radionov, V.F. et al. analysis or the current analysis. In particular, the discussion of the AOD over the Laptev Sea is not consistent between Table 2 and Figure 4. Table 2 shows the highest value (0.08 +/- 0.077) for all regions considered while the left side of Figure 4 shows that the Laptev Sea (LS) is lower than 3 out of 4 of the other regions. If part of the new analysis of the historical expeditions is associated with eliminating the impacts of forest fires on August 26, 2018 (Lines 239-240) then Table 2 should reflect this new filtering.
If the authors decide to keep the discussion of Figure 2 and Table 2, then the speculation regarding the reasons for the higher Vf and Vc in the Kara Sea should be further explored. For example, the authors speculate that “outflows of pollutants from oil and gas production plants in the north of the Western Siberia” (lines 216-219) may be the reason for the higher Vf and Vc (Figure 2) in the Kara Sea. From Figure 1, it appears that there is enough latitudinal sampling to construct estimates of North-South gradients (as done for the Barents Sea) for this region. Presenting this new information could be used to help support this speculation and help to explain the poor agreement with Mbc and Na values over the Kara Sea and extend the previously published work.
Specific Comments:
Figure 1 should include major anthropogenic emission sources since much of the discussion in the manuscript refers to potential sources of the observed aerosol distributions.
Line 49: “has few own sources” should be “has few local sources”. I also question this statement since there is extensive literature on biogenic marine aerosol measurements in the Arctic, for example:
Ferrero, L.; Sangiorgi, G.; Perrone, M.G.; Rizzi, C.; Cataldi, M.; Markuszewski, P.; Pakszys, P.; Makuch, P.; Petelski, T.; Becagli, S.; Traversi, R.; Bolzacchini, E.; Zielinski, T. Chemical Composition of Aerosol over the Arctic Ocean from Summer ARctic EXpedition (AREX) 2011–2012 Cruises: Ions, Amines, Elemental Carbon, Organic Matter, Polycyclic Aromatic Hydrocarbons, n-Alkanes, Metals, and Rare Earth Elements. Atmosphere 2019, 10, 54.
Line 86: “In 2019 we continued expedition studies of atmospheric aerosol over seas of the Arctic Ocean”. What about the previous nine expeditions? These should be summarized as well.
Line 151: how were “false measurements” identified? Please elaborate on the quality control procedures.
Line 203: Define acronyms for different Seas and Figure 2 caption (currently only defined in table 2)
Line 271-272: The statement that “These results indicate that the anthropogenic aerosol sources, located in Europe, are nevertheless predominant.” is inconsistent with the back trajectory analysis on April 22, 2019 (Figure 6) which shows that transport from “large oil and gas production centers” located along the coast of the Laptev Sea, well east of the 2019 Stage 1 measurements in the Barents Sea.
Line 273: Remove Russian text between “fine” and “black carbon”
Lines 396-399: "Back trajectory analysis of air mass motion gave no unique answer about what are the sources of ЕС and ОС and why their maxima occur in different times. In addition to the arrival of air masses from the Arctic Ocean basin, on April 7 and 11 there were also long–range transports of air from southwest of Siberia (Figure 11а).” Given that the measurements are over 2-3 day periods a single back-trajectory does not provide any useful information regarding airmass origins. An ensemble of back trajectories should be run throughout the measurement period to show the different source regions of the airmasses.
Line 399: “temperature anomalies” In what? Are you referring to satellite infrared fire detection? Please clarify.
References:
Radionov, V.F.; Kabanov, D.M.; Polkin, V.V.; Sakerin, S.M.; Izosimova, O.N. Aerosol characteristics over the Arctic seas of Eurasia: results of measurements in 2018 and average spatial distribution in the summer–autumn periods of 2007–2018. Problemy Arktiki i Antarktiki. Problems of Arctic and Antarctic. 2019, 4, 667 405–421. DOI: 10.30758/0555–2648–2019–65–4–405–421.
Reviewer 3 Report
Comments and suggestions are in the attached file.

Reviewer 4 Report
MDPI –Atmosphere Review
Spatial distribution of atmospheric aerosol physicochemical characteristics in Russian sector of the Arctic Ocean
Sergey M. Sakerin 1,*, Dmitry M. Kabanov 1, Valery I. Makarov 2, Viktor V. Pol’kin 1, Svetlana A. Popova 2, Olga V. Chankina 2, Anton O. Pochufarov 1, Vladimir F. Radionov 3 and Denis D., Rize 3
General comments:
The manuscript “Spatial distribution of atmospheric aerosol physicochemical characteristics in Russian sector of the Arctic Ocean” describes combined long-term distribution of AOD, BC and chemical composition of aerosols in Russian sector of the Arctic sea, mostly based on the new measurements from 2019 compared to previous year. This study is interesting and relevant for air quality, climate change issues, which in these times becomes extremely. This manuscript requires minor revision to be suitable for publication.
The main problem with the article is the purpose of the studies – it is not well defined in the introduction. It seems for me, that the author want to present really interesting results but do not have specified goals and ideas ”how to sell it”. Please, state the objective of yours work with proper background.
Specific comments about the substantive content:
The abstract should be discuss and rewritten again. I think it does not contain necessary information, just hard-given result data without any reference. Unfortunately, it does not tell me whether these values in this region are OK, are they too high or at least what they indicate.
In the abstract, I cannot find any information that these work are the results from one cruise compared to the results of the cruises from previous years. Please add this information.
Characterization of expedition measurements part should be rewritten as follows:
- Complete information about previous cruises – if you compare the data it is necessary to provide an information about date and time, number of measurement days etc.
- Changes in the Figure 1 provided in the pdf file.
The authors compare a really small dataset. I am not sure that those number of measurements are representative for the whole region. I wish that the authors will explain that in the text, also in the answers. For me, statistically is not enough for good paper, they should add any other data for comparison, even from a land collection or model results.
The authors are really consequent in building the text, paragraphs, preparing figures, making abbreviations and acronyms. It contains all required chapters, sufficiently developed, written used proper English with only several language mistakes - however, they do not affect the perception of work and analysis.
Returning to the topic of language, as a manuscript submitted for review, it is one of the few that is pleasant to read due to the fact that the authors put a lot of effort into making the content varied, avoid repetition, and construct sentences correctly. I must also emphasize that the method of analysis itself is correct, accurate and appropriately expanded, which, however, does not translate into the size of the data set used. This publication would be of great value if all the data used in this material appeared for the first time, has much more measurements, or - in this case, if they improved the reliability of the data, for example, with results from any atmospheric model as a supplement dataset.
The authors used 49 articles to cite in their work. Among these 49 there are 9 authors' works written by the first author, which constitutes about 20% of all positions, and 11 out of 49 positions are works by all authors of these manuscript. In my opinion, there are too many self-citations. Please change some of the self-cited works or add more works other than the authors of this manuscript in order to lower this statistic.
References are added correctly according to the Atmosphere style. Please, correct some mistakes highlighted in the pdf file.
